# With Great Sensitivity Comes Great Management: How Emotional Hypersensitivity Can Be the Superpower of Emotional Intelligence

**DOI:** 10.3390/jintelligence11100198

**Published:** 2023-10-11

**Authors:** Marina Fiori, Ashley K. Vesely-Maillefer, Maroussia Nicolet-Dit-Félix, Christelle Gillioz

**Affiliations:** 1Research and Development Division, Swiss Federal University for Vocational Education and Training (SFUVET), Avenue de Longemalle 1, 1015 Renens, Switzerland; 2Ecole Internationale de Genève, 1208 Genève, Switzerland

**Keywords:** EI_K_, EI_P_, emotional intelligence, hypersensitivity, emotion regulation

## Abstract

With the goal of furthering the understanding and investigation of emotional intelligence (EI), the present paper aims to address some of the characteristics that make EI a useful skill and, ultimately, a predictor of important life outcomes. Recently, the construct of hypersensitivity has been presented as one such necessary function, suggesting that high-EI individuals are more sensitive to emotions and emotional information than low-EI individuals. In this contribution, we aim to shift the perception of hypersensitivity, which is mostly seen with a negative connotation in the literature, to the perspective that hypersensitivity has the capacity to result in both negative and positive outcomes. We advance this possibility by discussing the characteristics that distinguish hypersensitive individuals who are also emotionally intelligent from those who are not. Based on an emotion information processing approach, we posit that emotional intelligence stems from the ability to manage one’s level of hypersensitivity: high-EI individuals are those who are better able to use hypersensitivity as an adaptive rather than a disabling feature. Ultimately, we propose that hypersensitivity can represent a sort of “superpower” that, when paired with regulatory processes that balance this hypersensitivity, characterizes the functioning of high-EI individuals and accounts for the positive outcomes reported in the literature.

## 1. Introduction

With just over 30 years of research, emotional intelligence (EI) has been catalyzing interventions in various domains and stands strong as a cutting-edge topic in research. Despite being a young domain of research, remarkable progress has been made in the latest decades to advance its progression in fundamental issues related to its conceptualization with respect to cognate constructs, its measurement, and its role in predicting important life outcomes. The debate surrounding the legitimacy of EI as a new scientific construct—quite fierce at times—has contributed to raising the quality of contributions and has guided scholars to address the most compelling issues EI has been confronted with ([13]).

In this article, we refer to the ([51]) conceptualization of ability EI as the expression of intelligence applied to the emotional realm of the individual by way of four interrelated facets: how individuals recognize emotions in oneself and others, how they use them to facilitate thinking, and how individuals understand and manage emotions in oneself and others. With the goal of further advancing the discernment and investigation into EI and its related constructs, the present contribution aims to address some of the characteristics that render EI a useful skill and ultimately a predictor of important life outcomes. Recently, the construct of hypersensitivity has been presented as being one such necessary function ([19], [20]). This idea was introduced in the context of the “hypersensitivity hypothesis”, which states that individuals with high EI are more sensitive to emotions and emotion information than individuals with low EI.

We aim to shift the perception of hypersensitivity, mostly seen in the literature as having a negative connotation, to the perspective that hypersensitivity has the capacity to result in both negative and positive outcomes. We advance this possibility by discussing the characteristics that distinguish hypersensitive individuals who are also emotionally intelligent from those who are not. The idea is that EI stems from the ability to manage one’s level of hypersensitivity: high-EI individuals are those who are better able to use hypersensitivity as an adaptive feature instead of an impairing characteristic. Ultimately, we posit that hypersensitivity represents a sort of “superpower” that, when paired with the capacity to balance this hypersensitivity with regulatory processes, characterizes the functioning of high-EI individuals and accounts for the positive outcomes known in the literature.

In guiding the reader through our understanding of how EI would function by way of hypersensitivity, we follow the chronological line of reasoning we developed over the last few years, which includes key articles that helped us to shape the theory presented in this contribution. We start by discussing the puzzling findings regarding “side effects” of EI, to further advance potential explanations rooted in how EI is operationalized, and the processes through which it comes into play. We then advance the hypothesis that hypersensitivity to emotion and emotion information is a fundamental feature of high EI, one that describes its *modus operandi*. We provide a definition and theoretical framework that can be used to this purpose, which includes the role of regulatory processes as a key function in the management of this hypersensitivity. We conclude by discussing open questions and future directions.

## 2. The Starting Point: Does EI Really Have a Dark Side? We Do Not Believe So

Our reasoning originated from analyzing the mounting evidence that EI might also have undesirable consequences ([15]; [19]). Effects at the intrapersonal level, such as the association between EI and depression or suicidal ideation in university students ([11]), or higher cortisol levels in stress situations ([5]; [38]), have been cited. These findings are puzzling first and foremost because (ability) EI should be an asset, not an impediment. Indeed, pairing the word “intelligence” with “emotional” implies that there should be adaptative benefits for individuals using emotions to support thinking. The term coined “wise mind” by [35] ([35], [36]), a very successful skills training module from dialectical behavior therapy (used for clinical purposes), supports this notion, emphasizing that effective decision making should neither ignore emotions using only logic (“reasonable mind”) nor rely only on emotions without accessing reason/logic (“emotion mind”). “Wise mind” allows for the integration of the two, resulting in more fulfilled choices that promote effective action. It allows one to honor and nurture emotions while also acting rationally, as opposed to, for example, suppressing feelings (reasonable mind alone) or reacting quickly or defensively (emotion mind alone).

The definition of EI as an ability, measured with maximum performance tasks, underscores the fundamental characteristic of EI as a form of intelligence. High-EI individuals should be *skilled* with emotions, which means that they should be able to outperform others in emotionally-connotated tasks (e.g., emotion detection) or in those situations charged with a heavy emotional load (e.g., interpersonal conflict, caring for a terminally ill loved one). If this does not appear to be the case, then we are facing a conceptual conundrum that requires an explanation. 

### 2.1. Hypersensitivity as a Possible Explanation of the Side Effects of EI

Recent attempts to explain the drawback effects of EI have suggested that if one is higher on EI, for instance, good at perceiving emotional stimuli, this might ultimately result in being more submerged by emotions, especially if the stimuli are negative (see also [2]). This idea was tested in a lab experiment by [19] ([19]), who showed that high-EI individuals (in particular, those high on emotion perception) were more strongly affected by induced incidental anger. They reported stronger anger reactions and provided more negatively biased ratings of an ambiguous target. A follow-up of this study ([20]) introduced the “hypersensitivity hypothesis” as a potential explanation for the notion that high-EI individuals amplified the importance of *both* positive and negative information when forming impressions of others. According to this hypothesis, individuals high in EI may be more sensitive to emotions and to emotion information than individuals low in EI. In this view, EI can be conceived as a magnifier through which individuals perceive and process the emotional aspects of their inner and outer lives. All in all, these results raise the issue that high-EI individuals are also hypersensitive, in that they experience an amplification of the valence and intensity of emotions, which then has an impact on (more or less advantageous) behavioral outcomes.

### 2.2. The Way EI Is Operationalized and Measured May Further Explain Negative Consequences of EI

Another reason why negative consequences might appear to arise in association with EI is because of different limitations in how ability EI is operationalized and measured in several studies. First, although EI is defined as being composed of different facets, studies rarely consider all of these facets when assessing EI. They often focus on an overall score, or on scores related to one or the other facet (see also [15]; [31]). As with all psychological concepts, if the tools used to measure it are missing or fail to characterize its components, one risks misrepresenting the construct and, thus, the impact it is deemed to have on important outcomes. In the cases noted above, though an individual may yield an overall high EI, scores may be made up by very high emotion perception and very poor emotion management, among other permutations. A case in point is the study by [11] ([11]), who made the specific link between increased depression, suicidal ideation, and hopelessness with *only* the emotion perception factor of EI measured as an ability. Further results of this study include individuals who scored lower on the factor of managing the emotions of others (another *part* of EI, but measured through self-report scale) who had greater suicidal ideation. These findings highlight that when only one facet of EI is considered—in this case regarding EI measured as an ability—the individual’s complete EI profile is not known, possibly explaining why the profile is associated with negative outcomes: the necessary components that render EI to be “skillful” could be missing. 

Second, as discussed in the EI literature, there is still a question about the extent to which current ability EI tests can capture EI *in action*, rather than mostly just knowledge about emotions. This would lead to a potential gap between scores on ability EI and observed behavior in context. A common feature of ability EI tests across EI facets is the requirement of deep reasoning about emotions; for instance, situational judgment tests typically engage test-takers into “if–then” conditional reasoning. Scores derived from such tests may be helpful to quantify the repertory of actions related to emotional situations; however, they may not fully account for how individuals would act themselves (as compared to a hypothetical character in a situation). They may additionally fail to account for the extent to which individuals would be able to engage in effortful thinking if they do not have access to their full cognitive or temporal resources (e.g., if they are submerged by a heavy emotional load or subtle emotional signs in real life). Though the cognitive reasoning piece around emotions does contribute to one’s EI, it was argued that the theorization and measurement of ability EI must also consider more automatic processing of emotion information ([18]), such as those relying on implicit methods within research in personality psychology ([48]). The below section (Section 2.3) makes suggestions around how using the proposed additions to EI measurement can lead us to better understand the connections between EI and varied outcomes, as well as to understand the relationship between hypersensitivity and EI (Section 3).

### 2.3. EI_P_ and EI_K_ Can Help Us Understand How EI Can Lead to Both Positive and Negative Outcomes

One way to address the limitations of current ability EI measures was the introduction of a new EI component representing how individuals experience emotions, such as how they respond to and process emotions and emotion information ([21]). The basic idea is that within a broad conceptualization of ability EI as a unique construct, two distinct components can be identified: EI_K_ and EI_P_. The first, EI_K_, or the emotion knowledge component, can be measured with current ability EI tests, such as the situational test of emotion understanding (STEU; [37]) or, in the workplace, with the Geneva Emotional Competence Test (GECo; [53]). The type of reasoning mainly involved in EI_K_ is top-down, wherein individuals start from general principles about how one should behave emotionally, and then contextualize to more specific situations and constraints. The second, EI_P_, or the emotion–information processing component, is a new component that can be measured with cognitive tasks assessing the efficiency of cognitive and emotional processing. This is typically performed by way of reaction time or rapid visual identification. This type of processing is bottom-up, based mainly on the sensorial properties of the stimuli, such as their saliency or intensity. Factor analyses conducted on the EI_P_ and EI_K_ components within the nomological network of intelligence show that the two components are correlated with each other, although the best fitting model is the one in which they are conceptualized as separate factors related to both fluid and crystallized intelligence ([21]). Hence, the two components should be thought of as related (within a unique EI factor), but also distinct from each other. 

The advantage of using this distinction is that it accounts, among other things, for a dual-process account of EI ([18]), as well as for more cognitive approaches, such as system 1 and 2 proposed by [30] ([30]). In particular, EI_P_ involves rapid and more instinctive emotional reactions and emotional contagion, presenting similarities to system 1; individuals may use this process to quickly respond to emotional cues in the environment. EI_K_ involves the conscious and deliberate evaluation and regulation of emotions, presenting similarities to system 2. Individuals may use thoughtful strategies and reasoning to recognize, understand, and manage emotions in oneself and others. 

Of note, individuals who qualify as having “high EI” should be high in both EI_K_ and EI_P_, with each component modulating the other in a homeostatic balance. In the absence of this balance, we might still incur negative outcomes, such as difficulties with social interactions, typical of individuals with Asperger’s syndrome (now labeled high-functioning autism). These individuals depend on high EI_K_, as they are often able to hold a good level of emotion knowledge (as measured by the MSCEIT), especially when given enough time to think through the options ([41]). Despite this, these individuals are partly characterized by difficulties with social interactions ([1]), appearing to have low EI_P_, which may account for a dearth in the perception of social cues. Without the processing of social cues, the cognitive understanding of them is less useful in actual social interactions. Other negative outcomes due to the imbalance between EI_K_ and EI_P_ are those typically associated with hypersensitivity effects, such as very intense and prolonged negative reactions, which may result from low EI_K_ (more specifically, low emotion management, which is part of EI_K_) and high EI_P_. 

Hence, the distinction between EI_P_ and EI_K_ helps us to better understand how EI would exert positive vs. negative effects, because it deepens the comprehension of what an unbalanced profile is. This includes disequilibrium not only among ability EI facets (or EI_K_), but also between EI_P_ and EI_K_.

## 3. Bringing It All Together: Emotion–Information Processing as the Theoretical Framework of Hypersensitivity

Up to this point we have advanced the possibility that what distinguishes hypersensitive individuals who are also emotionally intelligent from those who are not resides in their capacity to balance the “hypersensitive” function of EI with regulatory processes that would allow them to retain only the benefits of that hypersensitivity. We further push our understanding of the relationship between EI and hypersensitivity by referring to the notion that both EI_P_ and EI_K_ are required to accurately represent EI. This implies that they both play a role in determining the way hypersensitivity can be defined and the consequences it may have. 

To this purpose, we employ an information processing approach that integrates basic cognitive models (such as [4]) with emotion process frameworks (e.g., [17]) to conceptualize the function of hypersensitivity within ability EI. An information processing account of hypersensitivity describes hypersensitive individuals as those who have high levels of EI_P_, reflected in a lower threshold for the perception of emotional features, the ability to detect very subtle and fine-grained affective responses in oneself and others, attention directed preferably towards emotional stimuli, the experience of more intense emotional reactions to emotional stimuli at both the physiological and subjective level, and stronger memory for emotion-related information. In Figure 1 we present the different processes involved in the treatment of emotion information (orange boxes), which correspond to the EI_P_ component of EI and that may be considered the building blocks of hypersensitivity. We may find, for example, that hypersensitivity unfolds at the level of attentional mechanisms, with high-EI individuals being more attentive to emotional, as compared to neutral, pictures. Another possibility is that hypersensitive reactions might be generalized to all different information processing steps, i.e., all the different processes included in the boxes “input processing” and “further elaboration and storage” in Figure 1.

Investigating the specific processes involved in hypersensitivity in high-EI individuals may help to identify which ones are associated with the production of appropriate, adaptive responses, such as resilience or wellbeing (green box). Of note, the distinction between the color of the boxes highlights that hypersensitivity, which corresponds to high level of EI_P_ (orange boxes), describes the *modus operandi* of EI, rather than the *effects* of this way of operating, which relate to the *consequence* of hypersensitivity (green box). Importantly, as discussed earlier, the hypersensitivity of high-EI individuals is “managed” by emotion regulation, which is related to EI_K_ and is deemed to be the characteristic that renders their hypersensitivity an advantage (thus categorizing them as emotionally intelligent individuals). 

In sum, we claim that the interplay of EI_P_ and EI_K_—in particular, the EI aspect related to emotion regulation (ER)—ensures the adaptive functioning of hypersensitivity and characterizes high-EI individuals. Our theorization points out that EI stems from the interplay of these two components, with both having to be high in order to ensure adaptive outcomes. Indeed, the stronger the emotional reactivity of high-EI individuals, the more effective ER needs to be. Ultimately, we claim that hypersensitivity in high-EI individual functions like a superpower that requires the power to control such hypersensitivity through regulatory processes in order to ensure adaptive functioning. 

To further understand what this theoretical concept may look like in reality, we use a practical example to illustrate how a balance between EI_P_ and ER can be reached, and, thus, high EI reached, or not in Table 1. 

### Hypersensitivity to Pleasant and Unpleasant Emotions and Positive Outcomes 

A more complete picture of EI is emerging when considering the distinction between EI_P_ and EI_K_, *and* the hypersensitivity hypothesis, where high-EI individuals are those who possess a) enhanced processing of emotion and emotional information (hypersensitivity) and b) the ability to balance the “hypersensitive” function with regulatory processes. These processes allow him or her to maximize the benefits of hypersensitivity without being overcome by its disadvantages (the intelligence part of EI). Our theorization fundamentally disputes the idea that hypersensitivity has a uniquely negative connotation, introducing a perspective in which hypersensitivity has the capacity to produce both negative and positive effects. 

At the outset, the idea to pair EI with hypersensitivity may seem counterintuitive. There is a vast amount of research showing adverse effects of intense emotional reactions in response to (mainly) negative events. In the clinical literature, affect intensity is considered a form of dysregulation associated with various types of psychopathologies ([29]), such as several mood and anxiety disorders ([39]).

On the other end of the spectrum, research showing a positive association between stronger reactivity and wellbeing, and resilience in healthy individuals has started to emerge (e.g., [52]; [60]). More specifically, the concept of emotional flexibility, which concerns the capacity to adapt intensity and duration of emotional reactions to pleasant and unpleasant situations/stimuli ([60]), presents similarities with the basic skills constituting EI. For this reason, this concept of emotional flexibility might be relevant when describing the hypersensitivity in high-EI individuals. Emotional flexibility encompasses three key elements. Emotional awareness involves being in touch with and aware of one’s emotions, recognizing and labeling them accurately. Emotional acceptance focuses on the nonjudgmental acceptance of one’s emotions, whether pleasant or unpleasant; it involves recognizing and validating emotional experiences without trying to suppress or avoid them. Emotional adaptability refers to the ability to regulate and flexibly modify emotional responses to the demands of a given situation; it implies the ability to adjust emotional reactions appropriately and adaptively, taking the context and the goals one is trying to achieve into account. Overall, the concept of emotional flexibility implies that intense emotions, managed and experienced fully in an adaptive context, are an advantage (and an EI skill). In fact, emotional flexibility was found to be associated with higher trait resilience ([60]). 

This is also aligned with the research showing that more intense emotional reactions to both pleasant and unpleasant images were associated with higher wellbeing ([33]). More specifically, the procedure employed to distinguish reaction intensity (or peak intensity) from reaction duration (magnitude of the reaction) highlighted that reacting with intense emotions may have positive outcomes: it was the peak intensity in response to emotion-eliciting pictures that was associated with wellbeing and adaptive choice. Although EI was not taken into consideration in these studies, they show the most beneficial side of emotional sensitivity as linked to adaptive and functional behavior. It is precisely this side that should be at play for highly emotionally intelligent individuals, as compared to the disadvantageous side associated with psychopathology.

Further, research in the field of positive psychology has shown that people who flourish display greater positive emotional reactivity in response to pleasant events such as helping, playing, and interacting ([9]). This hypersensitivity to positive stimuli might play a key role in broadening the scope of attention and in noticing things to savor ([7]). Several studies have shown that a greater ability to savor positive experiences in one’s life leads to several positive outcomes, such as an enhancement of happiness ([8]), life satisfaction ([55]), and resilience and wellbeing ([54]). This enhanced ability to experience positive emotions might allow individuals to overcome negativity in the wake of negative events and to thrive in personal growth ([22]). 

Additional support for the idea that sensitivity to emotional stimuli may be beneficial under certain circumstances comes from evolutionary and developmental psychology, in particular the concept of “differential susceptibility”, proposed by ([6]). This theory suggests that people vary in their susceptibility or sensitivity to environmental influences, and this sensitivity can manifest in both positive and negative ways. Challenging the notion that some people are simply "vulnerable" to negative influences (such as stress or adversity) while others are "resilient", it theorizes that those who are more sensitive to negative influences may also be more responsive to positive influences. For example, supportive relationships or interventions would be expected to have a stronger impact among sensitive individuals, leading to more adaptive outcomes. Such considerations highlight the role of individual differences in developmental plasticity as a fundamental feature of environmental adaptation. 

Experiencing unpleasant emotions intensely, though less intuitive in terms of its positive impact at first glance, can lead to several benefits when this hyper-reactivity to negative emotions is managed well. For instance, in the field of educational studies, it has been suggested that reactivity to negative emotions is adaptive because it might enhance learning and achievement ([50]). This is also related to the fact that emotional reactions to negative events can improve cognitive processing. Stronger emotional response to negative images has been associated with better memory consolidation for those events compared to those with weaker emotional responses ([32]). Empirical studies in which stress was manipulated also show that activation of threat-related stimuli may help to mobilize resources and help one to finally cope effectively with perceived “danger” ([21]; [40]). An extreme example of this can be seen with Navy SEAL warfighters, who reacted more intensely to threatening stimuli than men who are not part of the Navy SEALs ([44]). This, of course, is adaptive in a war setting, where this intense reaction increases the chance of survival; however, it is acknowledged that (stronger levels of)/different techniques of emotion regulation will then be required to manage this intense reaction in a less hostile setting. This notion also highlights the fact that different levels of sensitivity may need to be regulated to different extents (or in different ways), depending on the context.

As pleasant emotions tend to be societally more appropriate in most social contexts (at least in Western cultures), they also tend to lead to less problematic outcomes. Clinically speaking, the repression of emotions (or the act of not experiencing (often negative) emotions) for longer periods of time has shown to be harmful psychologically and thus to result in reduced wellbeing outcomes and increases in psychopathology ([10]). There is value in being able to hold back intense emotions in order to reduce maladaptive reactions, such as managing one’s sadness (after a loss, for example) during work hours; however, the long-term suppression of intense emotions such as grief, for instance, most often results in negative consequences (e.g., [43]). Thus, it can be said that emotionally intelligent individuals harvest the benefits of their hypersensitivity to both positive and negative emotions and are characterized by an enhanced affective reactivity to positive and negative events subsequently sustained by up regulatory processes.

Hence, it is not the valence of emotions that determines whether the outcome will be positive or negative. It is, rather, fully experiencing intense emotions and channeling the correct action depending on the context. This important function is often what separates those with positive from those with negative outcomes. The ability to regulate emotions (such as appropriate expression versus acceptance of the emotion depending on context; appropriate duration of the emotion, as noted in the above examples, etc.) plays a fundamental role in this process. When individuals are not able to regulate their intense emotions, hypersensitivity leads to negative outcomes and different pathologies (see above and Table 1). Referring to the theoretical explanation of individuals with borderline personality disorder (BPD), the individual experiences a heightened sensitivity to emotional stimuli, while experiencing these emotions quickly and intensely (often with a long reaction duration), and a slow return to baseline ([34]; [35]). The disordered aspects are not a result of the initial perception or intensity of feeling, but the inability to inhibit or reduce the reaction as well as the difficulty of returning to a more stable emotional state (e.g., [34]). On the opposite, emotionally intelligent individuals, who, by definition, regulate their emotions adequately, may take full advantage of their hypersensitivity: they fully experience emotions and use this hyped function to obtain a deeper and more detailed apperception of the inner and outer world, without being negatively impacted by this way of functioning. Following from this, a distinction can be made when speaking about hypersensitivity and its link to psychopathology (and, thus, implying lower EI) versus hypersensitivity linked to thriving (and, thus, linked to higher EI).

## 4. Open Questions and Future Directions

This contribution attempts to integrate different lines of research developed during the last few years, each highlighting new perspectives on EI. The emerging overall picture provides an in-depth understanding of the processes through which EI may lead to positive outcomes. We provide an interpretative key of the EI functioning, and leave open several questions. Below, we summarize a few of these questions and share some insights about how they could be addressed in future research.

### 4.1. How Is Hypersensitivity Related to Sensory Processing Sensitivity?

When speaking about hypersensitivity from an information processing perspective, it is imperative that we consider its link to sensory processing sensitivity (SPS; [3]), a term greatly cited in the clinical literature, and that we acknowledge the similarities and differences. 

The definition we provide of hypersensitivity, which encompasses depth of processing, greater emotional reactivity, and acute awareness of subtle stimuli, presents some similarities with SPS. However, there are several differences with respect to [3] ([3]) theory: first, the current definition of hypersensitivity pertains only to emotional, rather than physical or environmental, stimulation. On the one hand, we have not yet been able to empirically verify ourselves whether emotional hypersensitivity and sensory hypersensitivity respond to the same underlying mechanisms. On the other hand, we also think that a hypersensitive person’s senses are not more developed than a nonhypersensitive person; it is, rather, their brain that perceives and processes more in-depth information, especially of an emotional nature. This may lead to a hyperactivity of the nervous system that can activate the person experiencing it. In the end, we think that hypersensitivity might depend on the amplification of emotional processing associated with sensory perception rather than on a more developed sensory perception. 

Second, sensory processing sensitivity is typically measured with self-report scales, whereas emotional (hyper)sensitivity associated with EI is measured through objective, performance-based tasks. Self-report hypersensitivity questionnaires typically ask people to position themselves on items describing typical hypersensitive indicators, such as “I am very sensitive to pain”. This way of measuring/estimating hypersensitivity is based very much on self-knowledge and the recognition of how one usually behaves. The framework of hypersensitivity as it is presented in this contribution, and in the empirical work we have been performing so far on hypersensitivity and EI (e.g., [23], [24]; [42]), relies more directly on how individuals react to emotions and emotional stimuli. For example, we present emotional facial expressions and test whether hypersensitive individuals pay more attention to them as compared to neutral facial expressions, or we employ very subtle and complex blends of expressions that only hypersensitive people can recognize. This way of measuring hypersensitivity is more intuitive and unconscious, and based on behavioral indicators, such as accuracy or speed of response to typically emotional stimuli.

Third, we introduce a “special case” of hypersensitive individuals: those who are both hypersensitive *and* capable of managing such hypersensitivity, namely, emotionally intelligent individuals. This subcategory of individuals is characterized by the fact that they perceive reality (the internal and external world) through a magnifying lens that makes emotional features more salient and impactful. Hence, such individuals have much more emotional information regarding themselves, others, and the external world that may in principle represent an asset with respect to individuals who do not possess this hypersensitivity. Ultimately, we claim that all high-EI individuals are hypersensitive, but not all hypersensitive individuals are emotionally intelligent.

### 4.2. What Is the Role of Emotion Regulation and What Is Its Relationship with EI_P_ and EI_K_?

Emotion regulation (ER) refers to the processes through which individuals influence which emotions they experience, when and how they experience them, and how they express them ([26]; [27]). This framework is well suited to understanding our conceptualization of hypersensitivity. It highlights important parameters that may affect the unfolding of hypersensitivity, such as the modification of the intensity, duration, or type of emotional response to better cope with internal and external demands. In the following, we explore the association between ER and each EI component: EI_K_ and EI_P_.

#### ER and the EI Components

Several theories of emotion regulation, such as the process model of emotion regulation ([25]) and the emotion regulation theory ([58]), emphasize the importance of awareness of emotions in the regulation process. The component of emotional awareness within these frameworks is, in our view, fundamental to managing hypersensitivity. These theories suggest that people engage in a variety of strategies to manage their emotions, such as cognitive reappraisal or expressive suppression, but these strategies are only effective if people are aware of their emotions (a top-down process) in the first place. For example, someone unaware of their anger may not manage the associated behavior urge and may not even know which emotion they need to be working on in order to change their behavior. Thinking about thinking (or metacognition) as well as thinking about feeling (or meta-affect) is, indeed, emerging as a key factor for self-regulation and emotionally intelligent behavior in several recent theorizations ([12]; [57]). 

Studies have found that individuals with higher EI (currently measured as EI_K_) tend to have better emotion regulation skills. [61] ([61]), for example, investigated the relationship between EI and the use of specific emotion regulation strategies when regulating others’ emotions; individuals higher in EI, especially emotion management, used more high-engagement strategies, such as perspective-taking and problem-solving, and fewer low-engagement strategies, such as suppression and avoidance, when regulating others’ emotions compared to individuals lower in EI. These findings support our hypothesis and suggest that high-EI individuals are those who can use effective ER strategies when regulating others’ emotions. Overall, it seems as if by developing their emotion management skills, an important facet of EI, individuals can improve their ability to regulate their own emotions and help others manage theirs. This then leads to better mental health and interpersonal outcomes. In addition, findings suggest that the emotion management facet of EI is the most strictly related to ER. 

In sum, studies and conceptualizations explained above provide various examples of the way in which a balance among ability EI facets is necessary in order to be “truly” emotionally intelligent. Evidently, numerous questions remain as to how this balance can be achieved. In this paper, we propose the inclusion of sufficient emotion regulation in order to manage high levels of hypersensitivity; are there other ways in which overall emotional intelligence can be achieved? For example, do all components of EI_K_ have to be sufficiently high? Or is there a means of compensating? We discussed how high emotion perception, for example, cannot stand without emotion regulation; however, could other facets help with the emotion regulation enough to compensate for a high-perception–low-management combination, or is the former a must? How does emotion understanding (the EI facet, more closely related to the key construct of emotional awareness discussed earlier) fit in? Another approach of looking into these combinations would be by considering scores of the different EI facets within persons, such as using latent profile analysis (for examples, see [31]; [47]), or by testing interactions between different EI components, such as emotion perception and emotion management. Though the latter approach is little developed in research, it has the potential to enlighten how the different EI components may work together.

Another aspect of our theorization that needs further development concerns the relationship between ER and EI_P_. Going back to several years ago, Davidson argued that “…regulatory processes are an intrinsic part of emotional behaviour and rarely does an emotion get generated in the absence of recruiting associated regulatory processes. For this reason, it is often conceptually difficult to distinguish sharply between where an emotion ends and regulation begins” ([14]). This is an essential consideration as, relatedly, it may not be easy to empirically disentangle emotional hypersensitivity or high-EI_P_ from processes involved in regulating it. This is the reason why, in Figure 1, ER is connected with the different emotion information processing steps with a dotted line. This point relates to the question around the separation between EI_P_ and EI_K_, with the main inquiry being how they are intertwined. For example, could there be physical processes that physically inhibit coping in certain cases (e.g., where a coping strategy like a grounding exercise would not work) due to biological or neurological mechanisms? How and to what extent can we look into this?

### 4.3. Why Does the Emotion Management Facet of EI Not Have a More Prominent Role in Our Theorization (as Summarized in Figure 1)?

We believe that the EI_K_ component, in particular emotion management, may not fully account for all aspects of emotion regulation because of the following open questions: Are the measurement issues, mentioned above, limiting the predictive power of the emotion management facet of EI (e.g., does emotion management capture how people actually react in emotional situations)?Are current ability EI tests that measure emotion management truly measuring the *ability* high-EI individuals have to regulate emotions? This question comes from the empirical observation (in our own studies as well as in other publications) that the emotion management facet of EI does not have much predictive power with respect to other EI branches, such as emotion understanding, even when outcomes imply a key theoretical role of emotion regulation/management (for an example, see [21]). Might it be a challenge to measure emotion management though performance tests? For example, the emotion regulation subtest of the GECo ([53]) is more related to personality than to intelligence.Whereas EI and its emotion management facet describe the capacities people have, emotion regulation captures their behavioral outcome, such as the strategies people use to manage emotions ([16]); hence the two are not equivalent.

Overall, the broader conceptualization of regulatory processes, namely, emotion regulation, instead of the more narrowly defined emotion management facet of EI, better characterizes our theorization of EI in relation to hypersensitivity. Importantly, the link between EI_K_ and emotion regulation ensures intelligent emotion regulation or attention to both the processes underlying ER as well as individual differences in how such processes may be employed for better outcomes ([45]); intelligent emotion regulation may be conceived as a flexible emotion regulation that takes into considerations various parameters, such as personal goals, personal characteristics, and situational factors; it presents similarities with the concept of emotional flexibility we discussed earlier in the manuscript.

### 4.4. Does Hypersensitivity Start Having Negative Consequences When the Level Is Extremely High?

Following from Table 1, in which we theorize how different levels of hypersensitivity EI_P_ might be related to different levels of EI, the question around any person’s capacity to manage hypersensitivity at extremely high levels arises. As theoretically, the emotion regulation required to transform extremely high levels of hypersensitivity would be “gigantic”, is this an indicator that most individuals with such hypersensitivity levels would fall within clinical case levels and hence have negative outcomes associated with hypersensitivity? This lies within the same line of thinking as the Yerkes–Dodson law (despite its variations in conceptualization and limitations) in which an inverted U-shaped curve is used to illustrate the relationship between stress/arousal and performance, with peak performance being reached with medium levels of arousal (e.g., [56]). It is possible that a certain amount of hypersensitivity might be helpful for EI and, thus, for positive outcomes, but too much might be likely to hurt. This is in line with the example of high-hypersensitivity EI_P_ (hypersensitivity), low ER in Table 1. For instance, an intense onslaught of frustration and self-doubt in response to confused facial expressions from a group of students listening to one’s lecture could result not only in an intense rumination process, but in a negativity “spiral” that could move into processes that inhibit coping mechanisms and thus lead to the inability to continue the lecture.

The question of whether too much hypersensitivity could still be too much remains an empirical and theoretical open question. From the one side, it might be that as long as emotion regulation is strong enough to manage any level of hypersensitivity, only positive outcomes would be expected. From a practical point of view though, it might be the case that very intense emotional reactions would be hardly managed, resulting in the “too-much-of-a-good-thing-is-still-too-much” effect ([46]).

### 4.5. Would Hypersensitivity Refer Only to Ability EI or Also to Trait EI?

Hypersensitivity is framed within an information processing approach, and research has shown that associations between EI and “hot” and “cool” cognitive processes were found to be stronger for ability EI and inconsistent for trait EI ([28]). Considering this, hypersensitivity, as defined in the current contribution, would theoretically associate more strongly with the conceptualization of EI as an ability, rather than a personality trait. However, some of the literature points to potential implications of trait EI in emotion–information processing/hypersensitivity. In our studies we have also found some evidence that hypersensitivity measured with performance tasks is associated with higher trait EI, although the effects observed are less consistent than with ability EI. This once again brings up the question around EI measurement as trait EI is measured via self-report and thus requires some self-awareness around one’s emotional capacities. Perhaps hypersensitive individuals, having higher attunement to emotional stimuli, would be quite accurate in the self-report of their hypersensitivity? Or, could they have high perception of emotions without necessarily having the remaining complimentary high-EI facets as well? These questions require further investigation.

## 5. Implications for Applied Research and Training

Our theorization regarding the role of emotional hypersensitivity in EI by way of ER has important implications for training, public policy, and assessment. The presented model emphasizes the significance of emotion regulation (ER) in transforming emotional hypersensitivity from a possible hindrance into an asset, marking it as a key characteristic of highly emotionally intelligent individuals. In order to improve overall EI, individuals could improve their management of intense emotional responses by training their ER skills. This structured approach has been shown to foster emotional resilience, enhance relationships, and improve overall wellbeing. EI training promotes self-awareness, aiding individuals to recognize triggers to heightened emotional responses. It also helps to challenge and reframe negative thought patterns, reducing emotional reaction intensity by altering their interpretations of situation. Further EI training practices based on the development of ER skills promote staying in the present moment, curbing rumination about the past or future (e.g., mindfulness), as well as help to manage physiological aspects of hypersensitivity (e.g., relaxation techniques). 

By investing in the improvement of EI skills, individuals with emotional hypersensitivity can lead more fulfilling lives, gaining greater management of their emotions, and thus engaging in more emotionally intelligent behavior. This shift in the conceptualization of hypersensitivity as no longer having a uniquely negative connotation proposes, in contrast, that hypersensitivity can be the source of EI, opening up new forms of (emotional) diversity. In addition, it reinforces the utility of training emotional competences (e.g., [59]), including those related to the management of hypersensitivity at school and in the workplace. 

In terms of measurement, our novel conceptualization of hypersensitivity disrupts the conventional belief that it can be accurately gauged solely through self-report questionnaires. Instead, we advocate for a paradigm shift towards the use of objective measures rooted in performance-based assessments of hypersensitivity within emotional tasks. We believe that the reliance on self-reports alone may inflate the prevalence of hypersensitivity, especially among those with low ER skills. This occurs because most questionnaires employ criteria that hinge on participants acknowledging the overwhelming intensity of their emotional reactions, a criterion that could potentially overstate the issue.

In contrast, performance-based tests offer a more nuanced and balanced view of hypersensitivity. These assessments can delve into the nuanced facets of hypersensitivity, uncovering characteristics such as lower threshold of perception of emotional stimuli or fine-grained discrimination of complex emotional stimuli. In the literature, there are a few examples of tasks that could be used for this purpose, although they require further validation. These include tasks such as the facial expressions blends (FEB), which requires identifying the emotions expressed in a series of morphed images created by blending on the same face two emotions expressed by the same person Gillioz et al. 2023b). Another task that might be employed to detect emotional hypersensitivity is the Dynamic Affect Reactivity Task (DART), a task designed to identify the precise moments of emotion onset, peak, and the speed of emotional fluctuations in response to emotional images ([49]).

## 6. Concluding Remarks 

In this paper, we advance the idea that EI is greatly influenced by one’s level of sensitivity to emotional stimuli, proposing that high EI relies on hypersensitivity. However, hypersensitive individuals require the ability to manage that hypersensitivity to be considered emotionally “intelligent”. In other words, people with high emotional intelligence are those who are more sensitive to emotion and are able to manage their hypersensitivity, using it as an adaptive, rather than a detrimental, characteristic. We characterize this way of functioning of EI with the analogy of an “emotional superpower” when this hypersensitivity is accompanied by the capacity to use it in the service of adaptive behavior. 

Despite our conviction that EI should be associated with positive outcomes, we leave open the possibility of minor hitches related to the use of hypersensitivity as a superpower. We suspect that the great management evoked in the title, necessary to render hypersensitivity a superpower, might deplete emotionally intelligent individuals (physically and mentally) and thus limit the frequency and/or duration of their abilities. Further research may clarify this point. The solution we see in this potential limitation is that emotionally intelligent individuals may know when to limit the use of their superpower; in other words, they may acknowledge when they are running out of resources and either take a break or restrain their regulatory capacity to situations that really deserve it. 

## Figures and Tables

**Figure 1 jintelligence-11-00198-f001:**
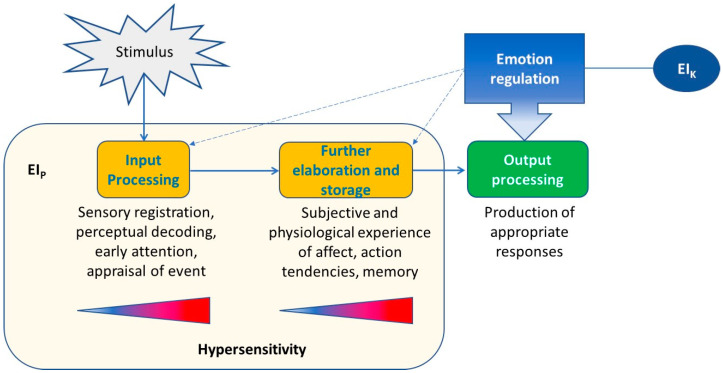
Description of the steps (orange boxes) leading emotionally intelligent individuals to achieve positive outcomes (green box), namely, emotional hypersensitivity plus the capacity to regulate such hypersensitivity. Levels of hypersensitivity are associated with EI_P_ in that the individual stands along a continuum from low (blue) to high (red) sensitivity based on different information processing steps, with hypersensitivity characterizing the functioning of high-EI individuals.

**Table 1 jintelligence-11-00198-t001:** High levels of EI_P_ (hypersensitivity) plus effective emotion regulation results in high EI. This table illustrates how hypersensitivity in response to an emotional event conceptualized as involving the cognitive processes shown in Figure 1 can lead to a well-managed versus poorly managed situation. When EI_P_ is high, there is a larger range possibility of EI. As long as the EI_P_ part is high, high EI is also possible but is not a given; this depends on the capacity to manage the feelings that result from this hypersensitivity. Given the intensity of emotion and reactivity this can cause, this may also lead to very low levels of overall EI (“in action”), as the consequences of not managing such hypersensitivity can be remarkable. On the other end of the spectrum, if EI_P_ is low, then high EI is not possible as the individual may not perceive emotional stimuli in the first place and thus would be less emotionally impacted by them.

	Situation: Giving a lecture in a classroom in which students are showing signs of boredom, annoyance, and/or difficulty understanding concepts through means such as furrowing eyebrows, rolling their eyes, worried expressions, giggling (reason unknown). The table below shows how different levels of EI_P_ are related to different levels of ER.		
	**High Levels of Emotion Regulation ^1^**	**Medium Levels of Emotion Regulation**	**Low Levels of Emotion Regulation**
**High levels of EI_P_ (hypersensitivity)**	**Possible Experience:** Intense feelings of frustration in response to student behaviors (e.g., not being able to meet the students’ needs/wants). **Possible Regulation Strategy:** Quickly and effectively uses positive self-talk to note that it is not personal; uses breathing techniques to calm the body down; thinks about challenging situations from the past in which he/she has been able to manage successfully. **Possible Highly Emotionally Intelligent Outcome:** Channels cognitive resources towards engaging students in their teaching (e.g., shifts tone of voice, gives an added example, or engaging students in discussion in order to shift the dynamic); lecture ends with great satisfaction of students and the teacher.	**Possible Experience:** Intense feelings of frustration at student behaviors (e.g., at not being able to meet the students’ needs/wants). **Possible Regulation Strategy:** Struggles to get through some breathing and positive self-talk, though still emotionally overwhelmed in the moment. **Possible Medium Emotionally Intelligent Outcome:** Gets through the lecture and then improves the content/examples for future lectures. Lecture ends with students being unclear about some of the content taught and the teacher being mildly frustrated by the situation experienced in class.	**Possible Experience:** Intense feelings of frustration at student behaviors (e.g., not being able to meet the students’ needs). **Possible Regulation Strategy:** Paralysis of any known coping strategies—student reactions taken as an attack on presenter’s lecturing ability. **Possible Non-Emotionally Intelligent Outcome:** Becoming overwhelmed and unable to move forward with lecture—bursting into tears, leaving the room, or screaming at the class to sit still and listen. Lecture is over without having covered all the content planned, teacher reputation is shifted, emotional exhaustion ensues.
**Low levels of EI_P_**	**N/A** Individual does not have the sensitivity to detect the relevance of student behaviors in connection with his/her teaching (high EI not possible).	**N/A** Individual may not notice the behaviors themselves and may not link them to his/her teaching, but maybe to the subject matter or other. The teacher thinks s/he is doing OK when in fact students are not interested in the lecture. **Possible Regulation strategy:** Teacher may make small attempts in voice tempo or tone in order to make sure students are fully attentive and engaged in the lecture. **Possible Non-Emotionally Intelligent Outcome:** Lecture ends with most students being unclear about content and bored by the teaching style. The teacher does not realize that his/her way of teaching is ineffective.	**Possible Experience:** The relevance of student behaviors to the lecture goes unnoticed. **Possible Regulation strategy:** None needed. **Possible Non-Emotionally Intelligent Outcome:** No change in lecture format or presentation. Students unsatisfied and bored, teacher does not question her/his way of teaching.

^1^ The levels of EI_P_ and ER are each divided into levels for the purpose of example. It is recognized that each of these lies on a continuum.

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
