# Peer review of "With Great Sensitivity Comes Great Management: How Emotional Hypersensitivity Can Be the Superpower of Emotional Intelligence"

_jintelligence, 2023, doi:10.3390/jintelligence11100198_

Round 1

Reviewer 1 Report

Overall, I think this is a potentially useful contribution to the special issue, though I do think the authors could make the manuscript more relevant to the readership, and theory development, by considering a few conceptual issues, as well attending to some issues of style.

Conceptual Issues

The authors talk at one point of an “emotional mind” and a “reasonable mind”. This is not an especially well articulated idea in the literature to my understanding, and it glosses over something that the authors could spend some time thinking through. Indeed, it possibly could lead to a better articulated theory. And that is this notion: The distinction between System 1 and System 2 thinking, which is considered an especially important distinction in the literature on cognitive processes (e.g., Kahneman, 2011). I suspect it could even account for the difference between high EI individuals and low EI individuals. In this model, using System 2 thinking in emotionally charged situation is the hallmark of those with high EI (see examples in Table 1 of the manuscript, which seem to cry out that this lens be applied as an explanatory mechanism). It seems, for mine, a plausible rival theory (notably without any recourse to hypersensitivity) that should be explored.

The more one thinks through this line of argument the more one comes to appreciate that while hypersensitivity might be a concomitant of emotional intelligence it does lead easily to a proper causal (i.e., explanatory) model. More likely candidates appear those tied to System 2 thinking, metacognition, and meta-affect (see e.g., Thomas et al., 2022). Even if the authors could make a more compelling case for their postulation, they should spend some time discussing this possibility. The way I see it, it is only through a third mechanism – like meta-cognition or meta-affect -- that highly emotional intelligent individuals might use their proposed hypersensitivity in adaptive ways.

Unrelated to the above, I am not certain why there is some discussion on the dark side of EI. It assumes there is no dark side to high forms of cognitive ability (a question that has perennially led to research papers, with equivocal findings). And in any event, dark side claims are based on a handful of studies with non-representative samples using limited instrumentation, and to my knowledge, have yet to be supported by any meta-analytic investigation. It seems more likely that a third variable, often unmeasured in the cited studies, leads to the findings. I would use these methodological problems to suggest the jury is out on the topic, and certainly not in any way suggestive of a dark side.

While I realize the authors have discussed their approach to measuring emotional hypersensitivity in another article, I also could not help but think this could benefit also from a more detailed exposition in the current instance, perhaps even sample items given as supplemental materials. I also suspect that whether hypersensitivity is related to either ability or trait EI will depend on how hypersensitivity is measured. If via self-report (e.g., I am very sensitive to others emotional pain) rest assured it will share meaningful relations with trait EI, as measured by instruments such as the TEIque.

I do think it important also that the authors address something that is singularly important in modern science, and that is what they see as the implication(s) of this model for practical applications having real world consequences, whether that be training, assessment, or perhaps public policy. It is well and good to have a conceptual model, but if that is not fueling something a practitioner can use, I find the manuscript slightly less thought provoking.

Issues of Style

The title is catchy, but I am afraid it is misleading. The authors correctly remain guarded about the relationship between emotional sensitivity and emotional management, talking at one point about an inverted U-function (i.e., too great a sensitivity could impair management). And while I am a huge fan of the MCU, talking about emotional hypersensitivity as being a “superpower” is far too colloquial (and as noted earlier in my review, hypersensitivity is likely not the true “hero” of this story).

The manuscript is generally well written, but I could not help but note how few short sentences can be found. Indeed, there are a couple of longish paragraphs based on just one sentence. I suspect some of these long sentences (which end up reading as convoluted) could be parsed, which will ultimately lead to greater reader engagement.

References

Kahneman, D. (2011). Thinking, fast and slow. New York: Macmillan.

Thomas, A. K., Wulff, A. N., Landinez, D., & Bulevich, J. B. (2022). Thinking about thinking about thinking… & feeling: A model for metacognitive and meta‐affective processes in task engagement. Wiley Interdisciplinary Reviews: Cognitive Science13(6), e1618.

Comments in main review should be addressed.

Author Response

Thank you for submitting your special issue paper titled “With great sensitivity comes great management: How emotional hypersensitivity can be the superpower of emotional intelligence” (jintelligence-2505933). I am thankful to the two knowledgeable reviewers, who were intrigued by your lines of argument. Overall, it seems to me that the paper is advancing some new ways of thinking (that follow from previous research) and does a good job in doing so. Nonetheless, there are ways that the paper can be improved and/or there are questions for you to think about. I will start by paraphrasing some of the reviewers’ comments (their full comments are below):

We are grateful to the Editor and the Reviewers for their thoughtful consideration of our work and the high-quality feedback provided. All the issues raised by the editor and the reviews deserve attention and further reflection and we have tried our best to address them as thoroughly as possible in the manuscript, which we feel is now improved.

1) Reviewer 1 presents what could be considered an alternative model – namely, that EI involves the use of system 2 resources to think about system 1 processing. The reviewer also suggests that your model could need some type of meta-cognition or meta-affect mechanism that would select management elements attuned to the situation. I don’t think the reviewer is asking you to overhaul your framework. However, thinking about the comments could lead to further insights and/or tweaking of your arguments.

As you can see from our replies to Reviewer 1, we think that system 1 and system 2 are, in fact, already embedded in our theorization, in particular in the distinction between EIK and EIP. Please see our reply to Reviewer 1, comment n.1, for an explanation about how the two EI components might be related to system 1 and 2. We agree we had not explicitly mentioned this relationship in the MS and we therefore added a new paragraph regarding this point on p. 4, lines 173-180.

Regarding the suggestion that meta-cognition or meta-affect might work by selecting elements attuned to the situation, we believe this corresponds to aspects of emotion regulation, one of the important functions within the model. For example, one might choose to avoid looking at disturbing pictures (an emotion regulation strategy) in order not to feel unpleasant emotions (goal derived from meta-cognition). In the end we believe that the functions the Editor and Reviewer 1 allude to could be played out by ER as the third variable that can explain the link between EI and hypersensitivity.

2) Reviewer 1 recommends that you spend some time discussing implications of the model, such as those related to training, assessment, or public policy. (In addition, or alternatively, you could articulate some specific research directions that would be informative. These would probably be a bit more specific than is currently present toward the end of the paper.)

Thanks for this suggestion. We added a new section on this, section n.5 on pp. 13-14.

3) Both reviewers find the title to be somewhat dramatic (I think), but I like the title. The paper is meant to provoke thought and new directions and I think you have succeeded here.

We are glad that we are on the same page regarding the title. It is meant to provoke indeed.

4) Reviewer 2 suggests that whether any ability leads to positive or negative outcomes is likely to depend on the context. For example, a very intelligent student may find certain classes boring, underperforming in these contexts.

Please see our comment to Reviewer 2, point 3, regarding the role of context.

5) Reviewer 2 is not sure the word hypersensitivity fits. I also wonder about whether hypersensitivity is just “sensitivity”. Sensitivity can be good or bad, depending on a variety of factors, but hypersensitivity seems to allude to problems. I am not sure you need to change your terminology, but there is something to think about here.

As we frequently discussed among the co-authors, the term hypersensitivity can either indicate a trait by itself, varying from low to high, or it may describe high levels of the trait ‘sensitivity’. In both cases, it is a matter of label, not a substantial difference. After reading your and Reviewer 2’s comment on this point, we are convinced it is indeed clearer to use the word ‘hyper-sensitivity’ to indicate high levels of the trait sensitivity.  We have made a few adjustments and introduced a new sentence to clarify this point in the manuscript.

I will separate my comments into questions/recommendations and minor.

Questions/recommendations:

1) One literature you should consider is a recent literature on mindfulness and acceptance. Lindsay and Creswell have proposed that there are two mindfulness skills – attention and acceptance. Attention is like sensitivity or hypersensitivity in that it makes the person aware of valenced feelings, which can increase emotional reactivity. However, when attention is paired with acceptance (like your management mechanism), mindfulness becomes quite functional. Could add this to explain how hypersensitivity works.

Thank you for this – for us, this is a great example of how emotion regulation can work together with hypersensitivity to result in high EI (and thus leading to positive outcomes). This being said, we are hesitant to specify with this example as, in our model, attention to emotionally valenced stimuli is only one piece of the puzzle that makes up sensitivity (alongside other sensorial properties of the stimuli, such as intensity or detection; could be a feeling without attention). High levels of attention placed on a strong feeling of frustration, for example, could certainly be coped with by practicing acceptance of the existence of that feeling (as done in dialectical behaviour therapy); however, this acceptance in our case would potentially also be referring to high bodily intensity related to frustration. And, yes, this could also be coped with using the regulation technique of mindfulness, though we worry about complexifying the argument if we were to explain this in too much detail.

2) Another literature you might consider is not the Aron and Aron (1997) framework, but a more recent literature on sensitivity as a mechanism of temperament (it is often linked to lesser serotonin function; see Pluess, Belsky, others). The sensitive child is considered an “orchid” that is capable of great achievements, but only if it is properly cared for (e.g., by supportive parenting). This model would seem to overlap with yours to some extent.

It seems to us that the Aron & Aron literature is more known to practitioners, whereas the Belsky and Pluess to researchers. In any case we found your suggestion noteworthy and added a paragraph on “differential susceptibility” on p. 8, lines 332-343.

3) Among other things, your analysis would seem to suggest systematic interactions between perception and management (e.g., if perception is high, but management is low, that is problematic, but if both are high, this may be the best combination). Has anyone looked at interactions of this type?

We think this is an interesting possibility, which has been little explored in the literature. We added this possibility in the MS when mentioning the importance of having a balanced profile in the EI facets on p. 11, lines 473-476.

4) Your framework distinguishes EI-K versus EI-P. Are these forms of EI correlated or totally uncorrelated? I think you will want to make concrete statements of this type, preferably on the basis of research findings. For example, what is the correlation between fluid and crystalized intelligence and might this correlation be a reasonable expectation concerning the relationship between EI-K and EI-P?

The covariance of the two-factor solution in a factor analysis, with EIK and EIP, was found to be (-).62 (Fiori et al., 2022). The two components are overall mildly correlated with each other, although less than the typical correlation between fluid and crystallized intelligence. We added this information in the MS on p. 4, line 169. In addition, correlations may vary depending on the EI facet considered and the type of emotional processes involved. For example, in our studies we found that performance in an emotional Stroop task is more correlated with the EI facet emotion management, whereas performance in an emotional dot-probe task correlate more with the facet emotion understanding.

5) I was curious about your take on Asperger syndrome, which seems correct to me, but I think it would be good to include a citation for your suggestion that this condition is knowledge with some difficulties in emotional processing (and/or, more automatic forms of EI).

We have added a relevant citation (Montgomery et al., 2010) to support our point (p. 4, line 187).

6) To some extent, you seem to move away from EI-K quite a bit later in the paper. For example, you suggest that K-perception cannot tap the more fluid aspects you are talking about and the K-management branch falls short in capturing more dynamic operations as well. I was not entirely sure about this direction as it seems to me that standard ability tests are tapping emotional processing and situation-contingent operations to some extent. I guess I am just wondering – are you wanting to jettison knowledge tests? What, concretely, would the alternatives look like? For example, are there reasonable batteries that could be used to assess perceptual and management operations independent of the sorts of problems that K tests use?

You are right that standard ability EI tests do tap, to some extent, emotional processing, otherwise they could not be subsumed under one overall EI factor together with EIP. However, we believe that ability EI tests tap much more into thoughtful and conscious processing. EIK and EIP are meant to predict partially different aspects of performance (see also Fiori & Vesely-Maillefer, 2019) and empirical evidence somewhat supports this prediction (Fiori et al., 2022), although it is a novel approach and more research is needed to ascertain this point.

As far as it regards EIK tests, we are deeply appreciative of the progress ability EI testing has made in the last few years. We believe that knowledge about emotions has an important role in predicting emotionally intelligent performance; it is in fact included in our theorization, together with EIP. It seems to us that EIK tests may not capture more spontaneous emotional performance, in particular quick emotional reactions in response to emotional stimuli embedded in the situation. Our current and past critique of these tests is meant to encourage confrontation, debate, and ultimately advance research on a key issue in EI, namely, how to measure it. Though we believe that something is missing from EI knowledge tests, this does not mean that the parts they do include are not still useful.

 In our research we have been using emotion information processing tasks to measure EIP, such as the emotional Stroop task, the emotional dot-probe task, or the emotional GoNoGo. However, these tasks have been used in the experimental literature, and there is ongoing discussion regarding whether and how they could be used as individual differences measures. The main challenge—beyond test-retest reliability--resides in the fact that empirical and correlation research are based on opposite assumptions: tasks in experimental research try to minimize between-subject variability; however, this is precisely the source of variance individual differences’ researchers are interested in. Hence, when you research combining the two approaches you often feel like a tightrope walker: you want to be able to observe variation in performance as a function of the experimental conditions, but you also want to have enough variability between participants within each experimental condition, to be able to rank individuals based on performance. A very interesting discussion on this issue, including possible solutions, is the paper by Hedge and colleagues (Hedge et al., 2018).

Other alternatives we are aware of, refer to tests that increase the difficulty of tasks and reduce the time to provide an answer so as to capture more spontaneous responses. Examples we are aware of include the Test battery for measuring the perception and recognition of facial expressions (Wilhelm et al., 2014) for the facet of emotion recognition, as well as a modified version of these tests we recently introduced to capture accuracy in fine-grained discrimination of emotional expressions (Gillioz et al., 2023). Another alternative we became aware of recently, is the Dynamic Affect Reactivity Task (DART, Robinson et al., 2023), which in our view could be an excellent test of emotional hyper-sensitivity, as we also mentioned in the MS on p. 14, lines 636-639.

7) Another literature you might want to consider is the neuroscience literature concerned with cognitive control, emotion regulation, and self-regulation. There are components of the brain that summarize perception – such as the insula and the ACC – and there are components of the brain that do regulation – such as various components of the frontal cortex. The perceptual components are often sensitive to affect (e.g., there are ERP components, generated by the ACC, that seem to respond to aversive outcomes and/or the amygdala has been described in alarm-related terms). If these more perceptual components are not linked to the frontal lobes in functional ways (e.g., the ACC recruits the DLPFC or VMPFC), you might have cases of clinical significance. When these links work correctly, you have a sophisticated cybernetic system. Note that in these models, perceptual sensitivity is absolutely critical to effective self-regulation, but cannot accomplish self-regulation (e.g., see Kerns et al., 2004, in Science, Botvinick, or Teper/Inzlicht). In cybernetic terms, there needs to be a monitor (perception) as well as an operator (management). You might find models of this type useful to your theorizing.

We thank the editor for providing interesting and related literature for our paper. We are less familiar with the neuroscience literature, and upon searching the authors mentioned in the comment, we were not able to find a publication that provides the editor’s argument indicating what happens when the perceptive brain component is not aligned with the regulatory component. This is indeed a very good point that would further support our argument on the interrelation between perception and regulation. We would appreciate a specific reference that would help us to cite this point in the MS; without a specific reference, we feel it is better not to broaden our discussion to include cybernetic models in order to avoid too much complexity, although we will certainly keep in mind your suggestion for further theoretical development.

8) I think the final considerations section could be a little bit tighter or more explicit.

We worked on this throughout the manuscript and rewrote part of the final considerations.

Minor:

1) p. 6. “…to an emotional event, conceptualized…Figure 1, can lead…”

2) I think Klein at al. is now 2023 and has page numbers.

3) p. 9. “…there are several differences with respect to the Aron and Aron (1997) theory…”

Thanks for noticing this, they are now fixed.

Conclusions

Your paper is very interesting. Please think about the comments above and below and prepare a revision along with a response letter. Thank you.

Sincerely,

Michael Robinson, Guest Editor

Email: Michael.D.Robinson@ndsu.edu

REVIEWER 1

Overall, I think this is a potentially useful contribution to the special issue, though I do think the authors could make the manuscript more relevant to the readership, and theory development, by considering a few conceptual issues, as well attending to some issues of style.

Conceptual Issues

1.The authors talk at one point of an “emotional mind” and a “reasonable mind”. This is not an especially well articulated idea in the literature to my understanding, and it glosses over something that the authors could spend some time thinking through. Indeed, it possibly could lead to a better articulated theory. And that is this notion: The distinction between System 1 and System 2 thinking, which is considered an especially important distinction in the literature on cognitive processes (e.g., Kahneman, 2011). I suspect it could even account for the difference between high EI individuals and low EI individuals. In this model, using System 2 thinking in emotionally charged situation is the hallmark of those with high EI (see examples in Table 1 of the manuscript, which seem to cry out that this lens be applied as an explanatory mechanism). It seems, for mine, a plausible rival theory (notably without any recourse to hypersensitivity) that should be explored.

The terminology of an “emotional mind” and a “reasonable mind” was simply used as a means to illustrate the practical applications of this type of thinking. It refers to a clinical skillset taught in order to promote thinking that includes both emotions and logic. We added a clarification to emphasize its clinical nature.

We do agree with Reviewer 1 that system 1 and system 2 play an important role for emotional intelligence (EI) and we are deeply supportive to approaches that have integrated this distinction into EI theorization (e.g., Fiori, 2009). However, instead of considering this distinction as a ‘rival’ theory, we consider it embedded in our present theorization of hyper-sensitivity. More specifically, hyper-sensitivity represents the EIP component, which concerns spontaneous, quick and mostly automatic processing of emotion information, presenting great similarities with system 1 of Kahneman theorization; as per our theorization, this component includes processes such as instantly recognizing complex emotional expressions, quickly reacting to threats, and having stronger physiological responses in response to others’ feelings. The EIK component, which is related to emotion regulation, instead, presents several similarities with system 2, including making complex emotional decisions, assessing the pertinence and appropriateness of more intuitive (or system 1) emotional reactions, and deciding how to respond to it.

In summary, we are grateful to Reviewer 1 for prompting us to make our understanding of EIP and EIK more explicit as related to hypersensitivity by way of System 1 and system 2: In essence, EI is a blend of both System 1 and System 2 processes. System 1 relates to the immediate emotional reactions and intuitive understanding of emotional context, while System 2 adds depth of processing by way of emotional awareness, empathy, and emotional regulation. Developing EI involves honing the skills of both thinking systems to enhance interpersonal relationships, navigate social situations, and make emotionally informed decisions.

We have added some of this thinking into the MS; in particular, we added a paragraph on p. 4, lines, pp. 173-180.

  1. The more one thinks through this line of argument the more one comes to appreciate that while hypersensitivity might be a concomitant of emotional intelligence it does lead easily to a proper causal (i.e., explanatory) model. More likely candidates appear those tied to System 2 thinking, metacognition, and meta-affect (see e.g., Thomas et al., 2022). Even if the authors could make a more compelling case for their postulation, they should spend some time discussing this possibility. The way I see it, it is only through a third mechanism – like meta-cognition or meta-affect -- that highly emotional intelligent individuals might use their proposed hypersensitivity in adaptive ways.

The third mechanism that high EI individuals can use to render hypersensitivity adaptive is, in our view, emotion regulation (ER). ER presents several similarities with meta-cognition: both processes require individuals to be aware of their mental states and to engage in self-reflection, although meta-cognition involves reflecting on one's own cognitive processes, whereas emotion regulation entails being mindful of emotional reactions. In addition, both processes involve self-regulation and internal monitoring as well as the engagement of executive functions. What distinguishes the two constructs the most is that meta-cognition pertains to cognitive processes and learning strategies, while emotion regulation focuses on managing emotions for adaptive functioning and for higher emotional well-being. We would like to mention that, when conceptualizing the model presented in the manuscript, we had a discussion regarding whether the place given to ER should have been taken by meta-cognition. In the end, given that the emotional component of cognition (a.k.a. emotional intelligence) was the key construct in the paper, we thought that ER was the best choice for the argument we wanted to make. Hence, although Reviewer’s 1 comments resonate with some of our thoughts, we still believe that ER deserves a more fundamental role in emotional intelligence as compared to meta-cognition.

Nevertheless, we thought it was a good idea to mention the link of emotion regulation with metacognition and meta-affect, and added a paragraph on p. 11, lines 473-476.

  1. Unrelated to the above, I am not certain why there is some discussion on the dark side of EI. It assumes there is no dark side to high forms of cognitive ability (a question that has perennially led to research papers, with equivocal findings). And in any event, dark side claims are based on a handful of studies with non-representative samples using limited instrumentation, and to my knowledge, have yet to be supported by any meta-analytic investigation. It seems more likely that a third variable, often unmeasured in the cited studies, leads to the findings. I would use these methodological problems to suggest the jury is out on the topic, and certainly not in any way suggestive of a dark side.

We agree with Reviewer 1 that current sparse findings regarding side effects of EI might be explained by a third variable, which in our theorization corresponds to emotion regulation. Despite not extensively supported by empirical evidence, there has been a discussion in the literature as well as in the EI community regarding potential side effects of EI. This discussion is also what prompted our reconsideration of the way of functioning of EI, and in our view it needs to be mentioned because it was the starting point of our reflection, as we also mentioned in the MS.

  1. While I realize the authors have discussed their approach to measuring emotional hypersensitivity in another article, I also could not help but think this could benefit also from a more detailed exposition in the current instance, perhaps even sample items given as supplemental materials. I also suspect that whether hypersensitivity is related to either ability or trait EI will depend on how hypersensitivity is measured. If via self-report (e.g., I am very sensitive to others emotional pain) rest assured it will share meaningful relations with trait EI, as measured by instruments such as the TEIque.

We agree it is important to clearly understand how each of the construct discussed are measured. We mention some possibilities on p. 10, second paragraph, regarding tasks used to look at the processing aspect of Emotional Intelligence (i.e., various levels of EIP), which then indicate levels of sensitivity to emotional information according to our conceptual model. We agree with the notion and are aware that the research shows that the means of measurement will greatly impact the level at which it will be related to either type of EI (or other construct measured through a lens of performance versus self-report for that matter). In the interest of not going into too much detail and taking away from the main purpose of the paper, we leave the discussion to section 5. Implications for applied research and training, in particular when discussing new ways of measuring EIP and hypersensitivity on p. 14, lines 619-639.

  1. I do think it important also that the authors address something that is singularly important in modern science, and that is what they see as the implication(s) of this model for practical applications having real world consequences, whether that be training, assessment, or perhaps public policy. It is well and good to have a conceptual model, but if that is not fueling something a practitioner can use, I find the manuscript slightly less thought provoking.

Reviewer 1 is right to encourage us to provide concrete examples of how our theorization can impact training, assessment and even public policy. We respond to this request by adding section 5. Implications for applied research and training.

Issues of Style

  1. The title is catchy, but I am afraid it is misleading. The authors correctly remain guarded about the relationship between emotional sensitivity and emotional management, talking at one point about an inverted U-function (i.e., too great a sensitivity could impair management). And while I am a huge fan of the MCU, talking about emotional hypersensitivity as being a “superpower” is far too colloquial (and as noted earlier in my review, hypersensitivity is likely not the true “hero” of this story).

We respectfully disagree with Reviewer 1 that hypersensitivity is not the hero of the story, as there would not be superpower without hypersensitivity. Although ER remains the key factor in our conceptualization (the third variable mentioned earlier by Reviewer 1), we are attached to the idea of hypersensitivity + emotion regulation being like a superpower, first and foremost because it challenges the view that hypersensitivity has uniquely negative consequences. In addition, we believe that the use of the analogy of EI as a ‘superpower’ constituted by hypersensitivity + ER may help to make the argumentation a bit ‘lighter’, given the complexity of the model proposed.

  1. The manuscript is generally well written, but I could not help but note how few short sentences can be found. Indeed, there are a couple of longish paragraphs based on just one sentence. I suspect some of these long sentences (which end up reading as convoluted) could be parsed, which will ultimately lead to greater reader engagement.

We thank for the comment, and we agree that we had several sentences that were too long. We revised the MS accordingly.

References

Kahneman, D. (2011). Thinking, fast and slow. New York: Macmillan.

Thomas, A. K., Wulff, A. N., Landinez, D., & Bulevich, J. B. (2022). Thinking about thinking about thinking… & feeling: A model for metacognitive and meta‐affective processes in task engagement. Wiley Interdisciplinary Reviews: Cognitive Science, 13(6), e1618.

REVIEWER 2

  1. The authors focus on hypersensitivity in the context of EI. The focal argument is that hypersensitivity can have both positive and negative effects. This is explained by reformulating the EI concept into two meta-abilities related to knowledge and processing. These are then placed below another meta-ability – emotion regulation in the widest sense. Overall, the authors make a very plausible point. Less use of hyperbole, such as talking about a “superpower” might further improve the perceived plausibility. However, I am wondering whether the authors are not starting from a point of complexity that is not quite necessary by ignoring the context dependence of successful EI use.

Specifically, the article sets out with a discussion on whether having too much of an ability can have negative effects. The authors argue against that notion on the basis of face validity. The main argument seems to be that one cannot have too much of a good thing. However, this argument ignores real life and specifically the context dependence of any ability. It is well known for example that highly intelligent kids can perform badly in school because they are bored due to lack of challenge. Similarly, one can imagine many contexts in which hypersensitivity has negative effects – as the authors outline later themselves. It seems this section only serves to set up a conundrum that does not exist and is not needed as an argument device.

We thank Reviewer 2 for acknowledging the plausibility of our argument. Regarding the section reminding of the too-much-of-a-good-thing effect, or section 2., this section was meant to describe how our line of reasoning developed around the concept of hypersensitivity. We followed a chronological order, and what first prompted our reflection was the debate about an alleged dark side of EI. This section is therefore fundamental for us to explain how we arrived to our conclusions.

For a discussion on the role of context, please see our reply to comment n. 3.

  1. In fact context dependence seems at the core of arguments throughout the text. For example, the authors provide an example for hypersensitivity discussing the heightened reactions of Navy SEALS to threat. Given the job profile of Navy SEALS seems to me that those who did not react with great sensitivity to threat might not have been available for study. In other words, in the context of being a Navy SEAL we are not so much talking about hypersensitivity but rather about necessary sensitivity. These individuals are only hypersensitive compared to participant populations without ongoing concerns about threat.

We acknowledge Reviewer’s comments, which we think is well taken; we adapted the Navy SEALS example to show that context matters on p. 9, lines 359-366.

  1. As such it seems to me that when taking context into account, much of the debate about hypersensitivity boils down to the fact that different environments present different challenges for which different levels of sensitivity are required. This then implies that whenever actual sensitivity does not match required sensitivity some sort of adaptation is required. In the case of too much sensitivity this may require a higher level of emotion regulation skill, in the case of not enough sensitivity this may require more emotion knowledge. Acknowledging this might make the argument less convoluted, while keeping the basic triad of EIK, EIP, and emotion regulation in the center of the argument.

We are thankful to Reviewer 2 for pushing our reflection further and for bringing into consideration the role of context. Overall, we consider plausible that the hypersensitivity inherent in all high EI individuals may need more or less regulation depending on the context of the person (…or maybe depending on the individual’s goals?). However, in our view the key role for adaptation is played by emotion regulation rather than the environment, meaning that the same exact context may produce very different reactions in individuals, which may become adaptive depending on the capacity to (down)regulate hypersensitivity. In other words, individuals with higher levels of sensitivity could either thrive or suffer in the same context – this is the main argument we are trying to get across.

Regarding the point: “In the case of too much sensitivity this may require a higher level of emotion regulation skill, in the case of not enough sensitivity this may require more emotion knowledge”, the case of not enough sensitivity mentioned by Reviewer 2 would not fall in a possible scenario involving high EI individuals, because in our theorization high EI individuals are by default hypersensitive. Medium levels of sensitivity can be compatible with high EI, and in such a case it seems like a good idea that EIK could complement sensitivity.

At a broader level, the explanation suggested by Reviewer 2 is definitely plausible and actually does fit within the framework and theorization we are proposing: in fact, the EIP component, which represents the way of functioning of hypersensitive individuals, is thought to be ‘bottom up’ processing influenced by the sensorial properties of the stimuli, ultimately being conceived as a contextualized component of EI.

Reviewer 2 Report

The authors focus on hypersensitivity in the context of EI. The focal argument is that hypersensitivity can have both positive and negative effects. This is explained by reformulating the EI concept into two meta-abilities related to knowledge and processing. These are then placed below another meta-ability – emotion regulation in the widest sense. Overall, the authors make a very plausible point. Less use of hyperbole, such as talking about a “superpower” might further improve the perceived plausibility. However, I am wondering whether the authors are not starting from a point of complexity that is not quite necessary by ignoring the context dependence of successful EI use.

Specifically, the article sets out with a discussion on whether having too much of an ability can have negative effects. The authors argue against that notion on the basis of face validity. The main argument seems to be that one cannot have too much of a good thing. However, this argument ignores real life and specifically the context dependence of any ability. It is well known for example that highly intelligent kids can perform badly in school because they are bored due to lack of challenge. Similarly, one can imagine many contexts in which hypersensitivity has negative effects – as the authors outline later themselves. It seems this section only serves to set up a conundrum that does not exist and is not needed as an argument device.

In fact context dependence seems at the core of arguments throughout the text. For example, the authors provide an example for hypersensitivity discussing the heightened reactions of Navy SEALS to threat. Given the job profile of Navy SEALS seems to me that those who did not react with great sensitivity to threat might not have been available for study. In other words, in the context of being a Navy SEAL we are not so much talking about hypersensitivity but rather about necessary sensitivity. These individuals are only hypersensitive compared to participant populations without ongoing concerns about threat.

As such it seems to me that when taking context into account, much of the debate about hypersensitivity boils down to the fact that different environments present different challenges for which different levels of sensitivity are required. This then implies that whenever actual sensitivity does not match required sensitivity some sort of adaptation is required. In the case of too much sensitivity this may require a higher level of emotion regulation skill, in the case of not enough sensitivity this may require more emotion knowledge. Acknowledging this might make the argument less convoluted, while keeping the basic triad of EIK, EIP, and emotion regulation in the center of the argument.

Author Response

(The authors gave the same response as above.)

Round 2

Reviewer 1 Report

The authors have done a good job addressing the major issues the two reviewers and editor have raised in the previous version. I see no reason to have them go through another round of review. I still have reservations about the title, the foray into the dark side of EI, and the status of emotional regulation in this model, but the action editor was assuaged, and for mine, represents the definitive opinion. Congratulations to the authors.

Author Response

We thank Reviewer 1 for acknowledging that the paper has been improved. We were intrigued by Reviewer’s 1 comments, and also acknowledge that true discussion around the issues raised would require a debate.  It is hard to explain each other’s positions going back and forth with the revisions. We appreciate the reviewer’s opinion about the title, though wish to keep it the way it is. We believe that it reflects what we, as the authors, convey with our work, while provoking thought – this is part of our aim. As explained in our previous response letter, we are attached to the idea of presenting hypersensitivity as a sort of superpower and we would like to keep this analogy in the title in order to fulfill this aim. We hope reviewers will ultimately accept our choice.

Reviewer 2 Report

My main concern regarded the use of the term hypersensitivity when instead they actually talk about strong (and typically context-adequate) reactions. The prefix hyper is connoted with excessive and excessive is by definition not adequate. The author's terminology is therefore confusing at best. However, they seem to like the term.

Author Response

As we also mention in the MS, we aim to shift the perception of hypersensitivity, mostly seen in the literature as having a negative connotation, to the perspective that hypersensitivity has the capacity to result in both negative and positive outcomes. We changed the word hypersensitivity to reflect high sensitivity. High sensitivity does not mean excessive in our conceptualization, although we raised the point that very strong levels of sensitivity might require very strong levels of emotion regulation to manage it (I.e., it can, but does not necessarily).